# A Semi-automated Instrument for Cellular Oxidative Potential Evaluation (SCOPE) of Water-soluble Extracts of Ambient Particulate Matter

Sudheer Salana, Yixiang Wang, Joseph V. Puthussery, Vishal Verma

Department of Civil and Environmental Engineering, University of Illinois at Urbana-Champaign, Urbana, 61801, USA

*Correspondence to*: Vishal Verma (vverma@illinois.edu)

**Abstract.** Several automated instruments exist to measure the acellular oxidative potential (OP) of ambient particulate matter (PM). However, cellular OP of the ambient PM is still measured manually, which severely limits the comparison between two

types of assays. Cellular assays could provide a more comprehensive assessment of the PM-induced oxidative stress, as they incorporate more biological processes involved in the PM-catalyzed reactive oxygen species (ROS) generation. Considering this need, we developed a first of its kind semi-automated instrument for measuring the cellular OP based on a macrophage ROS assay using rat alveolar macrophages. The instrument named SCOPE - Semi-automated instrument for Cellular Oxidative Potential Evaluation, uses dichlorofluorescein diacetate (DCFH-DA) as a probe to detect the OP of PM samples extracted in

water.  SCOPE is capable of analyzing a batch of six samples (including one negative and one positive control) in five hours and is equipped to operate continuously for 24-hours with minimal manual intervention after every batch of analysis, i.e., after every five hours. SCOPE has a high analytical precision as assessed from both positive controls and ambient PM samples (CoV <17 %). The results obtained from the instrument were in good agreement with manual measurements using tert-Butyl hydroperoxide (t-BOOH) as the positive control (slope = 0.83 for automated vs. manual, $R^2$ = 0.99) and ambient samples

(slope = 0.83, $R^2$ = 0.71). We further demonstrated the ability of SCOPE to analyze a large number of both ambient and laboratory samples, and developed a dataset on the intrinsic cellular OP of several compounds, such as metals, quinones, polycyclic aromatic hydrocarbons (PAHs) and inorganic salts, commonly known to be present in ambient PM. This dataset is potentially useful in future studies to apportion the contribution of key chemical species in the overall cellular OP of ambient PM.

**1 Introduction**

Epidemiological models have traditionally relied on mass of the particulate matter (PM) as a metric to associate the health effects such as wheeze (Doiron et al., 2017; Karakatsani et al., 2012), asthma (Holm et al., 2018; Wu et al., 2019; Zmirou et al., 2002), myocardial infarction and coronary heart disease (Yang et al., 2019), ischemic heart disease and dysrhythmias (Pope et al., 2004) and heart rate variability (Breitner et al., 2019; Pieters et al., 2012; Riojas-Rodriguez et al., 2006) with the

inhalation of ambient and indoor PM. However, mass is not a wholesome metric as it does not capture the diverse range of

particle physicochemical characteristics. Apparently, the assumption that an increase in PM mass alone leads to a proportionate increase in the mortality would yield erroneous estimates if we do not account for the complexity of PM chemical composition and the resulting intrinsic toxicities. There are also mounting evidence that toxic effects of different chemical components are not simply additive, but there exists both synergistic and antagonistic interactions (Wang et al., 2020; Yu et al., 2018).

Therefore, we need a metric of the PM along with mass that can provide some relevant information to assess its toxicity. Oxidative stress has emerged as one of such metrics, which has been identified as a crucial step in the progression of many human diseases.

Oxidative stress is caused by an imbalance between reactive oxygen species (ROS) generation and their subsequent scavenging by lung antioxidants (Kryston et al., 2011; Li et al., 2008; Møller et al., 2010; Rao et al., 2018; Reuter et al., 2010). Thus,

measuring the ability of PM to induce ROS generation in the respiratory system, also called the oxidative potential (OP), could be considered as one of the markers of its toxicity and accordingly several acellular assays have been developed in the recent past to measure the OP of PM. These include the dithiothreitol (DTT) assay (Charrier and Anastasio 2012; Fang et al., 2015), ascorbic acid (AA) assay (Künzli et al., 2006; Visentin et al., 2016), glutathione assay (Künzli et al., 2006; Mudway et al., 2005), hydroxyl radical ($^{\cdot}$OH) (Vidrio et al., 2009; Xiong et al., 2017) and electron paramagnetic resonance (EPR)

measurements (Dikalov et al., 2018; Jeong et al., 2016). Along the similar lines, several cellular assays have also been developed, which involve molecular probes that can detect ROS through their transformation from non-fluorescent to fluorescent forms (Dikalov and Harrison 2014; Kuznetsov et al., 2011; Landreman et al., 2008; Wan et al., 1993). However, measurement of OP of PM using both cellular and acellular assays is often a labor-intensive and time-consuming process and therefore manually analyzing a large number of ambient samples for spatiotemporal resolution of OP is a cumbersome process.

In the last few years, a number of automated instruments have been developed based on acellular assays which could provide rapid and high-throughput analyses of the PM chemical OP (Berg et al., 2020; Fang et al., 2015; Gao et al., 2017; Venkatachari and Hopke 2008). There have also been a number of online instruments which can be deployed in the field making it possible to collect real-time OP or ROS data (Brown et al., 2019; Huang et al., 2016; Puthussery et al., 2018; Sameenoi et al., 2012; Wragg et al., 2016; Zhou et al., 2018). Although, acellular assays have many advantages over cellular assays such as ease of

application, low maintenance and no risk of microbial contamination, they are unable to capture the complex biochemical reactions occurring in a biological system as a response to PM exposure. This could be one of the reasons for their inconsistent correlations with various biological responses such as DNA damage and expression of inflammatory cytokines in previous studies (Crobeddu et al., 2017; Janssen et al., 2015; Øvrevik 2019; Steenhof et al., 2011). Cellular assays have an edge in this regard as these assays directly expose biological cells to chemical constituents of the particles, thus capturing some, if not all,

of the biochemical processes related to the oxidative burst. However, to the best of our knowledge, no automated instrument has ever been developed to provide a rapid high-throughput analysis of the cellular OP induced by the ambient PM. In comparison to chemical assays, cellular assays are even more time and labor-intensive. Due to this strenuous nature of the

cellular protocols, there have been very limited comparison between the chemical and cellular OP measurements. There is a need for the development of an automated instrument for the cell-based measurement of OP, which could not only analyze a large number of samples in shorter period with minimal manual intervention, but could also open up the possibilities for developing a field-deployable real-time instrument measuring cellular OP. Having such an automated instrument would be able to provide a direct comparison of cellular and acellular assays, thus screening the important chemical OP endpoints. Such advances will also help in integrating the OP data in toxicological and/or epidemiological studies by yielding a relatively large dataset on these measurements.

In this paper, we describe the development of a semi-automated instrument for cellular oxidative potential evaluation (SCOPE). SCOPE is the first of its kind instrument to measure the cellular OP induced by the water-soluble ambient PM extracts in murine alveolar cell line NR8383, using an automated protocol. The instrument is capable of analyzing a batch of six samples (including one negative and one positive control) in five hours. SCOPE is equipped to operate continuously for 24-hours with minimal manual intervention after every batch of analysis, i.e., after every five hours. We also calculated the detection limit of this instrument and evaluated its performance by measuring precision and accuracy using both positive controls and ambient samples. Finally, we demonstrated the ability of SCOPE to analyze a large number of both ambient and laboratory samples, and developed a dataset on the intrinsic cellular OP of several compounds, such as metals, quinones, polycyclic aromatic hydrocarbons (PAHs) and inorganic salts, commonly known to be present in the ambient PM.

## 2 Materials and Methods

### 2.1 Chemicals

Copper (II) sulphate pentahydrate [Cu (II)] (≥98 %), Luperox® TBH70X, tert-Butyl hydro peroxide (t-BOOH) solution (70 wt. % in water), iron (II) sulphate heptahydrate [Fe(II)] (≥99 %), manganese (II) chloride tetrahydrate [Mn(II)] (≥98 %), zinc (II) nitrate hexahydrate [Zn(II)] (≥98 %), iron (III) chloride hexahydrate [Fe(III)] (≥97 %), lead (II) acetate trihydrate [Pb(II)] (≥98 %), aluminum (III) nitrate nonahydrate [Al(III)] (≥98 %), chromium(III) nitrate nonahydrate [Cr(III)] (≥97 %), cadmium (II) nitrate tetrahydrate [Cd(II)] (≥98 %), vanadium (III) chloride V(III) (97 %), nickel (II) chloride hexahydrate [Ni(II)] (99.9 %), 9,10-phenanthraquinone (PQN) (99 %), 1,2-naphthaquinone (1,2-NQN) (97 %), 1,4-naphthaquinone (1,4-NQN) (97 %), 5-hydroxyl-1,4-naphthaquinone (5-H-1,4-NQN) (97 %), pyrene (Pyr) (98 %), naphthalene (Naph) (99 %), anthracene (Anth) (97 %), phenanthrene (Phen) (98 %), benzo[a]pyrene (B[a]P) (≥96 %), fluorene (Flu) (98 %), benz[a]anthracene (B[a]A) (99 %), sodium chloride (NaCl) (≥99 %), ammonium nitrate ($NH_4NO_3$) (≥99 %), 2′,7′-dichlorofluorescin diacetate (DCFH-DA), zymosan A from saccharomyces cerevisiae, Ham's F-12K growth media, and fetal bovine serum (FBS) were purchased from Sigma Aldrich Co. (St. Louis, MO). Ammonium chloride ($NH_4Cl$) and calcium chloride ($CaCl_2$) were purchased from VWR Life Sciences. Ammonium sulphate ($NH_4SO_4$) and potassium chloride (KCl) were purchased from Fisher Scientific. Salt

glucose media (SGM) at two different concentrations (1X and 10X), which was prepared according to the composition discussed in Klein et al., (2002), was provided by the Cell Media Facility at UIUC.

## 2.2 Stock Solution Preparation

The stock solutions of quinones (PQN, 1,2-NQN and 1,4-NQN) were prepared in DMSO, stored in a freezer at -20 ℃ and used within a week. The stock solution of 5-H-1,4-NQN was prepared on the same day of the experiment as it was found to be relatively unstable (e.g., change in color over a period of more than 24 hours) compared to other quinones. The stock solutions of PAHs (anthracene, pyrene, naphthalene, fluorene, phenanthrene, Ba[A]P, and Ba[A]A) were prepared in methanol, stored in a freezer at -20 ℃ and used within a week. On the day of the experiment, the stock solutions of both quinones and PAHs were diluted using Milli-Q deionized water (DI, resistivity = 18.2 M$\Omega$/cm) to appropriate concentrations. A 45 mM stock solution of DCFH-DA was prepared and aliquoted into different vials (30 µL per vial). These vials were stored in a freezer (at -20 ℃) and used within a month. To prepare the final probe solution, a portion of the content of one vial (i.e., 25 µL of 45 mM DCFH-DA) was diluted 100 times just before the experiment, using 10X SGM. All the metals, inorganic salt and t-BOOH solutions were freshly prepared using DI on the day of the experiments and immediately used.

## 2.3 Cells

Alveolar macrophages form the front-line of defense in pulmonary region of respiratory system against attack by the foreign particles. These cells play a major role in preliminary responses such as phagocytosis, secreting pro-inflammatory cytokines and killing pathogens. We have used a murine cell line, NR8383, as it is one of the most widely used cell lines in the PM studies. Certain characteristics of this cell line make it one of the best macrophage models available for the evaluation of PM OP. These characteristics include minimal maintenance (can be studied in a BSL-1 lab) and highly reproducible results that are comparable to primary cells (Helmke et al., 1988). NR8383 also expresses a number of inflammatory cytokines such as IL-1$\beta$ and TNF-$\alpha$ (Lin et al., 2000), thus it will allow us to link the results obtained from this instrument to these inflammatory responses, in our future studies. The cells were maintained on glass culture plates in Ham's F12-K medium containing 5 % FBS and incubated at 37 ℃ with 5 % $CO_2$ concentration. The cells were cultured by transferring floating cells from culture plates to fresh plates every four weeks. The cells generally divide and double in concentration within 48 hours (Helmke et al., 1987), after which the floating cells are removed for further growth of the attached cells by adding fresh media. Since Ham's F-12K media could itself contribute to the fluorescence, it was replaced by 1X SGM after counting the initial cell density and subsequent centrifugation, such that final concentration of the cells in SGM is 2000 cells/µL.

Before designing the protocol of our instrument, we conducted an experiment by keeping the cells outside an incubator but in a temperature-controlled environment (i.e., 37 ℃ maintained through a thermomixer used in our instrument) and measured the cell viability using trypan blue [see Fig. S1 in the supplementary information (SI)]. We found that over a period of 5 hours,

the cell viability decreased by only 6 %. However, the cell viability started decreasing sharply beyond 6 hours. Therefore, we limited the cells exposure to the outside environment for only five hours.

## 2.4 System Setup

We adapted the method of macrophage ROS assay from Landreman et al., (2008) which is the most widely used protocol for measuring the cellular OP of ambient PM. In this assay, DCFH-DA is used as an ROS probe. The reaction mechanism of DCFH-DA with ROS is well established (Rosenkranz et al., 1992; Wan et al., 1993). Briefly, DCFH-DA is a cell permeable compound which undergoes deacetylation by intracellular esterase to form DCFH. DCFH is oxidized by a variety of ROS to form a fluorescent product called DCF. The intensity of fluorescence provides a direct measure of the ROS generation. We measured the variation in absolute fluorescence of DCFH-DA as a function of time to assess the possible degradation or autooxidation of DCFH-DA during our measurement. The results showed that the absolute fluorescence of DCFH-DA remains constant for a period of at least 6 hours, indicating the stability of the probe within our experimental timeframe (please refer to Fig. S2 in SI). In our protocol, all the components of the assay, i.e., sample (138 μL), molecular probe (DCFH-DA; 39 μL), and cells suspension (177 μL at a final concentration of 1000 cells per μL in the reaction vial (RV)) are added together and incubated for 2 hours. Next, a small aliquot is withdrawn and transferred to a spectrofluorometer after dilution to measure the fluorescence. The incubation time of 2 hours was chosen after measuring the kinetics of ROS generation for two PM samples (chosen randomly from the sample set analyzed in our study) at a time interval of 30 minutes over a 3.5 h time period (please refer to Fig. S3 in SI). It was found that the ROS response peaks and stabilizes at around 2-hour incubation time for both of the PM samples. These results are consistent with Landreman et al., (2008), which also reported that for most samples (PM, blanks, positive control), the ROS response stabilized at around 2-hour incubation time.

The schematic diagram of SCOPE based on this protocol is shown in Fig. 1. The instrument consists of four major units: cells reservoir and samples holder, fluid transfer unit, incubation-cum-reaction unit, and measurement unit. The cells reservoir and sample holder consist of a set of seven vials (15 mL each) – one containing NR8383 cells suspended in 1X SGM, one amber vial containing DCFH-DA solution, five vials containing samples (i.e., four PM samples and one positive control). All the vials of this unit were placed in an Eppendorf Thermo-Mixer (Eppendorf North America, Hauppauge, NY, USA), which is maintained at 37 °C while continuously shaking at a frequency of 600 RPM. The fluid transfer unit consists of three Kloehn programmable syringe pumps (IMI precision, Littleton, CO, USA) (Pump #1, 2, and 3; see Fig. 1) and a 14-port multi-position valve (VICI® Valco Instrument Co. Inc., Houston, TX, USA) connected to Pump #2. The incubation-cum-reaction unit consists of 17 RVs [amber vials, 2 mL each; 1 for negative control (i.e., the cells treated with DI) in triplicate, 1 for positive control (t-BOOH) in duplicate and 4 for PM samples in triplicates] held in another Eppendorf Thermo-Mixer which is maintained at 37 °C and continuously shaking at a frequency of 800 RPM to keep the contents of all the vials well-mixed and suspended. 14 of these RVs are connected to Pump # 2 through the multi-position valve. Each RV connected to the multi-position valve is accessed by changing the valve position [using a valve actuator (VICI®)] to its respective number. Since

multi-position valve has only 14 ports, rest 3 RVs are directly connected to Pump #3. Both of these pumps (i.e., #2 and 3) transfer the content from various reservoirs (e.g., cells, DI and DCFH-DA) to RVs, and also transfer a small aliquot from these RVs (50 µL from each RV) to the measurement vial (MV) after 2 hours of reaction. Finally, the measurement unit consists of a Fluoromax-4 spectrofluorometer (Horiba Scientific, Edison, NJ, USA) equipped with a Flowcell (Horiba Scientific, HPLC Flowcell- 25 µL volume) to measure the fluorescence generated from the reaction of DCFH and cellular ROS. Pump #1, which is connected to the MV and the spectrofluorometer, first dilutes the aliquot withdrawn from the RV and then transfers this diluted mixture from MV to the spectrofluorometer for fluorescence measurement.

## 2.5 Ambient PM Sample Collection and Preparation

Ambient $PM_{2.5}$ samples were used in this study for assessing the precision and accuracy of SCOPE. These ambient samples were collected as a part of the Midwest Sampling Campaign and the sampling procedure and collection protocol are described elsewhere (Yu et al., 2019). Briefly, $PM_{2.5}$ samples were collected on quartz filters (Pall Tissuquartz TM, 8"x10") using a high-volume sampler (flow rate of 1.13 m$^3$/min; $PM_{2.5}$ inlets, Tisch Environmental; Cleves, OH) from five different sites in the midwestern USA: a road-side site in Champaign (within the UIUC campus), a rural site in Bondville (IL), and three urban sites in Chicago, IL (university campus of Illinois Institute of Technology), Indianapolis, IN (Indiana University-Purdue University campus), and St. Louis, MO [a part of National Core Pollutants (NCore) Network of USEPA]. All the filters were prebaked at 550 ℃ for 24 hours before sampling. All the samples used in this study were collected between May 2018 and May 2019. A total of 50 samples from all the five sites (10 from Indianapolis, 9 from Chicago, 10 from St. Louis, 7 from Bondville and 14 from Champaign) were used for conducting the performance evaluation, i.e., assessing precision and accuracy of the instrument. Further details on these samples (i.e., dates of collection, exact mass loadings etc.) are provided in Table S1 of the SI.

## 2.6 Filters Extraction

A single circular section of 1 inch diameter was punched from the high-volume filter, immersed in DI and sonicated for 60 minutes in an ultrasonic water bath (Cole-Palmer, Vernon-Hills, IL, USA). The volume of DI was determined based on the $PM_{2.5}$ mass loading on each punched section, such that the final concentration of the extract for exposure in the RV is 30 µg/mL. After sonication, the extracts were passed through a 0.45 µm pore size polytetrafluoroethylene (PTFE) filter to remove any insoluble particles and/or filter fibers. The water-soluble $PM_{2.5}$ extracts were then used to measure the OP of the PM. Although sonication could potentially lead to the formation of ROS (Miljevic et al., 2014), we found that ROS response of a blank filter extracted in DI by sonication was only slightly higher than that of DI (average ratio of blank filter to DI = 1.17± 0.02; N= 20). Moreover, we always blank corrected the ROS response of a PM sample with that of the field blank filter. Therefore, any effect of sonication caused by the extraction of filter in water should have been largely cancelled out. We also assessed the impact of fluorescent particle smaller than 0.45 µm in our ambient PM. Specifically, we extracted 10 randomly chosen PM samples from the sample set analyzed in our study, extracted them in DI, filtered the extracts through a 0.45 µm

syringe filter, and measured their fluorescence at the same wavelengths (excitation 488 nm/ emission 530 nm) as used for
DCF. The difference between absolute fluorescence of the filtered extracts ($0.52 \pm 0.04$ fluorescence units) and DI ($0.47 \pm 0.1$ fluorescence units) was not statistically significant ($p > 0.05$; unpaired t-test). The absolute fluorescence of the filtered PM extract was 60-80 times lower than that of a negative control. Thus, the contribution of fluorescent ambient particles smaller than 0.45 µm to the ROS measurement is negligible.

## 2.7 OP Measurement Protocol

The protocol for measuring cellular OP involves two stages –the first stage consists of manual preparation of the cells, DCFH-DA probe, and PM extracts, while the second stage involves incubating the cells with PM and DCFH and measuring the fluorescence in an automated manner. After preparing the cells, DCFH-DA and different PM extracts (i.e., completion of first stage), all the vials are manually transferred to the cell reservoir and sample holder. The second stage (automation stage) further consists of two phases – reaction phase and measurement phase. The complete algorithm of 2nd stage is shown in Fig. 2. In
the reaction phase, various reactants (i.e., cells, DCFH-DA and PM extract or positive control of DI) are transferred from their respective reservoirs to the RVs using pump # 2 and 3. This is done in a sequence of steps: in the first step, 138 µL of DI is withdrawn using Pump #2 and transferred via the multi-position valve to three RVs marked for negative control (i.e., triplicate analysis of the negative control). In the second step, 177 µL cell solution is withdrawn from the cell reservoir using Pump # 2 and transferred sequentially to all 14 RVs via multi-position valve. Simultaneous to this step, 177 µL cell solution is withdrawn
from the cell reservoir using Pump # 3 and transferred to three RVs connected to that pump. In the third step, 39 µL DCFH-DA is transferred using pump #2 and pump #3 to the respective RVs connected to them (i.e., 14 RVs connected to pump #2 through the multi-position valve and 3 RVs directly connected to pump #3) following the same sequential order as for addition of the cell solution. Finally, 138 µL of positive control (t-BOOH) and PM extracts are transferred to the respective RVs using Pump #2 and # 3, i.e., t-BOOH and 3 PM extracts are transferred using Pump # 2 via multi-position valve, while one PM
extract using Pump # 3 directly connected to 3 RVs. After all the RVs are loaded with the reactants, SCOPE performs a single round of self-cleaning, in which all the valves and tubing of the instrument are rinsed with DI using the fluid handling unit (i.e., all three Kloehn pumps).

After 2 hours of cells' exposure to PM, the measurement phase starts in which the fluorescence of DCF formed in each RV is
measured in a sequential manner. Each measurement involves three steps- 1) withdrawing an aliquot of 50 µL from the RV (using Pump #2 for 14 RVs connected to it, and Pump #3 for the remaining 3 RVs) and transferring it to the MV; 2) diluting the aliquot 100 times by adding DI using Pump #1 to the MV, and finally 3) pushing the diluted aliquot through Flowcell of the spectrofluorometer using the same syringe Pump #1. The withdrawal of the aliquot from different RVs follows the same order as for their preparation, such that the cells in each vial undergo exposure to the PM extract or DI or t-BOOH for exactly
2 hours. The spectrofluorometer is preset at an excitation/emission wavelength of 488 nm/530 nm. Between successive fluorescence measurements of different RVs, the Flowcell, MV and the tubing connected to the multi-position valve are

thoroughly rinsed with at least 10 mL of DI. After all RVs are measured for fluorescence, the instrument performs a final round of thorough self-cleaning, wherein each valve and tubing are cleaned (three times) with 70 % ethanol followed by DI. All the RVs and MV after this cleaning step are disposed and replaced manually with clean empty vials. SCOPE takes about five hours for complete analysis of one batch of six samples (i.e., 4 PM extracts, one negative and one positive control). For the next batch of analysis, cells, DCFH-DA and samples are manually replaced with freshly prepared vials. In our experiments for this manuscript, one batch was run per day, although it is possible to run up to three batches (a total of twelve PM samples) per day.

## 3 Results and Discussions

### 3.1 Instrument Calibration

The results of OP of the samples (i.e., field blank filter, positive control or PM extract) are reported as the percentage increase in fluorescence relative to the negative control, which is consistent with many previous studies (Sun et al., 2011; Thayyullathil et al., 2008; Wan et al., 2012; Wang et al., 2012). Normalizing by the negative control which is analyzed in the same batch of the samples is important, because absolute fluorescence of the cells treated with negative control is not stable and vary in each experiment. Two factors could cause this variability in apparent response of the cells. First, DCFH-DA, being a photo-chemically active compound (Castro-Alférez et al., 2016; Chen et al., 2010), could itself undergo possible decay and slight photo-degradation over time. Second, the exposed cells could be under different developmental stages, which affects their metabolic activity and the subsequent generation of ROS. Both of these factors yield substantial variability [Coefficient of variation (CoV) = 35 %, as obtained from the experiments conducted on 20 different days] in absolute fluorescence of the cells treated with DI (see Fig. S4 in the SI). However, normalizing the fluorescence of a sample with that of the negative control minimizes this variability. For example, CoV for the ratio of the fluorescence caused by the positive controls (zymosan, concentration = 100 µg/mL) versus respective negative controls was only 16 % (Fig. S4). Therefore, fluorescence of all the samples (i.e., filter blank, field blank, positive control or PM extract) was normalized with that of the negative control, analyzed in the same batch of the samples. This normalized fluorescence of the sample was then blank corrected by subtracting corresponding fluorescence of the blank, which was DI for the positive control and field blank filter extract for the PM extract.

Fig. 3 shows the response curve for various concentrations of t-BOOH (3.51 – 87.83 mg/mL), which was used to calibrate the instrument. The calibration equation shown in Fig. 3 was used to convert the blank-corrected OP (% increase in fluorescence) to the equivalent units of mg/mL t-BOOH (see Sect. S1 in the SI for calculations). At concentrations higher than 87.83 mg/mL t-BOOH, the curve becomes non-linear (see Fig. S5 in the SI), but here we show only linear portion of the curve for the convenience of calculating the calibration equation.

### 3.2 Limit of Detection (LOD)

The LOD of SCOPE is defined as three times the standard deviation of multiple blanks. For this study, the LOD was calculated from the field blank filters (FB, N=10) analyzed in different batches. As discussed earlier, the OP response from these blanks was expressed as percentage increase in fluorescence with respect to corresponding negative control (analyzed in the same batch as FB). The LOD obtained from the average and standard deviation of this data is converted to equivalent units of t-BOOH (mg/mL) using the calibration equation shown in Fig. 3. The LOD obtained by this method is 1.26 mg/mL t-BOOH. Note, an ideal expression of LOD should have been in terms of the threshold PM mass required to yield a signal significantly distinguishable from the blanks, however, expressing LOD in those units is complicated given the variability associated with PM, such as different chemical composition of the PM samples collected on different days and the variability in extraction procedures (e.g., volume of water used for PM extraction and the filter area which can be submerged in that volume). Nevertheless, based on several experiments conducted in our lab, we found it is difficult to detect a signal for a PM extract with concentration below 20 µg/mL, which could be considered as a rough measure of the detection limit for SCOPE.

### 3.3 Precision and Accuracy

For determining analytical precision of SCOPE, three different types of samples, i.e., positive controls (9.75 mM t-BOOH and 100 µg/mL Zymosan), Cu (II) solution (20 µM), and water-soluble PM extracts were used. For PM extracts, ten different circular sections (each 1 inch in diameter) were punched from ten different spots on one of the high-volume filters collected in the Midwest Sampling Campaign (Sect. 2.5) and extracted in DI (Sect. 2.6). The blank corrected % normalized OP response of the PM extract was further normalized by the concentration of PM extract (30 µg/mL) used in the RV for exposure to the cells. Table 1 shows the average, standard deviation and CoV obtained from the measurements of various samples.

The instrument showed a CoV less than 20 % for most cases suggesting high reproducibility of the results. Among the positive controls, CoV for zymosan was the highest (39 %), as compared to 14 %, 14 % and 16 % obtained for Cu (II), PM samples and t-BOOH, respectively. We suspect that higher CoV for zymosan is partly due to water-insoluble nature of zymosan (Gao et al., 2012; Venkatachalam et al., 2020), which is often used as a suspended particle in phagocytosis assays (Sung et al., 1983; Thomas et al., 2007; Underhill, 2003). This could lead to deposition of zymosan particles inside tubing of the instrument, leading to an underestimation in the OP measurement. In contrast, t-BOOH is highly water-soluble [700,000 mg/L (OECD/SIDS, 1995)] and thus involves no such complications. t-BOOH is a well-established inducer of oxidative stress, not only in macrophages (Lopes et al., 2017; Prasad et al., 2007; Roux et al., 2019) but also in a variety of other cells such as hepatocytes (Kučera et al., 2014), sperm cells (Fatemi et al., 2012), and lung fibroblast cells (Lopes et al., 2017). t-BOOH diffuses through the cell membrane quite efficiently and has been demonstrated to induce a comprehensive oxidative stress response through the generation of a variety of species including $H_2O_2$, alkoxyl and peroxyl radicals. t-BOOH has also been found to be more stable in the cellular systems (Abe and Saito 1998), and also a better at glutathione (GSH) depletion (Dierickx

et al., 1999), inhibiting peroxiredoxin activity (Ikeda et al., 2011), evoke a more consistent cellular antioxidant response (Alia
et al., 2005), cause a greater DNA damage (Slamenova et al., 2013) and promote a more efficient peroxidation of membrane
lipids as compared to other oxidants such as $H_2O_2$ (Guidarelli et al., 1997). Our results along with these studies suggest that t-
BOOH could be a more reliable positive control than zymosan for the macrophage ROS assay, particularly for the automated
operation of our instrument.

The accuracy of SCOPE was evaluated by comparing the instrument's response with that obtained from the manual operation
using both positive controls and ambient PM samples. We prepared different concentrations of t-BOOH from 3.51 to 878.29
mg/mL (3.51,35.13,87.83, 175.66 and 878.29 mg/mL, in the RVs used for exposure to the macrophages), and analyses were
conducted both manually and using the instrument. Fig. 4 shows the comparison of manual and automated measurements of
ROS induced by various concentrations of t-BOOH. The slope of the automated versus manual measurements for the positive
control was ~0.83 with a very high coefficient of determination ($r^2$ = 0.99). The automated measurements were slightly but
consistently lower than the manual measurements. This bias could probably be caused by the error introduced during transfer
of cells using the fluid-transfer unit (i.e., some loss of cells in valves or tubing), leading to slight inconsistency of the cell
density in RVs. Though, this deposition of the cells is not expected to yield cross contamination of the samples, given a rigorous
cleaning procedure (as discussed in Sect. 2.7) employed during operation of the instrument.

One of the major objectives of developing SCOPE was to enable a high through-put analysis of the PM samples. To
demonstrate this ability of the instrument, fifty ambient $PM_{2.5}$ samples collected from various sites in the Midwest US (Sect.
2.5) were analyzed and the results from the automated instrument were compared with manual measurements. The results are
expressed in terms of the equivalent units of t-BOOH (mg of t-BOOH per mg of PM), and the comparison is shown in Fig. 5
(see Sect. S1 of the SI for the calculation procedure). Overall, there was very good comparison between the manual and
automated measurements, with a slope of 0.83 and a coefficient of determination ($r^2$) = 0.71.

**3.4 Intrinsic OP of individual PM chemical species**

To demonstrate the utility of SCOPE, we tested several compounds commonly known to be present in the ambient PM. These
include 11 metallic species [Fe (II), Fe (III), Cu(II), Mn(II), Zn (II), Al (III), Pb (II), Cr (III), Cd (II), V (III) and Ni (II)], 4
quinones (PQN, 1,4-NQN, 1,2-NQN and 5-H-1,4-NQN), 7 PAHs (Phenanthrene, Anthracene, Naphthalene, Pyrene, Fluorene,
B[a]P and B[a]A) and 6 inorganic salts (KCl, NaCl, $NH_4Cl$, $NH_4NO_3$, $NH_4SO_4$ and $CaCl_2$). The concentrations used for these
compounds, i.e., 0.5 μM for metals, 0.2 μM for quinones and PAHs, 5 μM for KCl, $NH_4Cl$, $NH_4NO_3$, $NH_4SO_4$, $CaCl_2$ and 1
μM for NaCl, were in their typical ranges present in the ambient $PM_{2.5}$ and similar to those used in previous studies based on
acellular assays (Charrier and Anastasio, 2012; Yu et al., 2018). We are not aware of any study which has systematically
explored and compared the DCFH-based OP of individual PAHs vs. various metals or quinones in alveolar macrophages
(murine cell line NR8383).

Fig. 6 shows the OP of these chemical species. To assess significant differences in the OP responses, we used a one-way ANOVA (analysis of variance) test followed by Tukey's test for post-hoc analysis on the intrinsic OP responses of different groups of the species, i.e., metals, organic and inorganic compounds. Among metals, Fe (II), Mn (II), and Cu (II) induced the highest response (9.95-12.40 mg/mL t-BOOH). Although, the OP of these three metals were not statistically different from each other, their responses were significantly different from the rest of the metals ($p < 0.05$). Other metals [Fe (III), Zn (II), Pb (II), Al (III), Cr (III), Cd (II) and V(III)] induced very low response (<4.5 mg/mL t-BOOH), and there was no statistical difference among their responses ($p > 0.05$). Interestingly, the pattern of Fe (III) vs. Fe (II) OP response (~3 times lower response of Fe III than Fe II) matches with their relative redox activities as measured by the dithiothreitol (DTT) assay, i.e., 3 times lower intrinsic DTT activity of Fe (III) compared to Fe (II) (Charrier and Anastasio, 2012).

Among the organic compounds, PQN and 1,2-NQN showed the highest response (7.51 and 6.52 mg/mL t-BOOH, respectively), however, their responses were significantly lower ($p > 0.05$) than that of the metals Fe (II), Mn (II) and Cu (II). Other than these two quinones, the OP of any of the organic compounds, i.e., PAHs, 1, 4-NQN and 5-H-1,4-NQN was not significantly above the negative control. PQN and 1,2-NQN are among the most abundantly found quinones in ambient air (Charrier and Anastasio, 2012), known to show a high redox cycling capability transitioning to and from their semiquinone forms, as well as the ability to cause DNA damage and induce apoptosis in cells (Klotz et al., 2014; Shang et al., 2014; Shinkai et al., 2012; Yang et al., 2018). Therefore, a high intrinsic OP of these quinones indicates towards their prominent role in other cellular responses such as inflammation and cell death. The insignificant contribution of PAHs in the cellular OP measured in our study is in contrast to several studies conducted on bronchial epithelial BEAS-2B cells (Landkocz et al., 2017), acute monocytic leukemia THP-1 cells (Den Hartigh et al., 2010) and U937 cell line (Tsai et al., 2012), which have suggested that PAHs such as B[a]A, B[a]P, pyrene, anthracene and phenanthrene are the important drivers of oxidative stress and cytotoxicity. However, these cells of human origin differ significantly from the murine cell lines used in our study in terms of their morphology (Krombach et al., 1997), expression of certain reactive nitrogen species and related enzymes (Jesch et al., 1997), and membrane proteins (Jaguin et al., 2013). Certain mechanisms, such as aryl hydrocarbon receptor (AhR)-mediated activity which activates the CYP450 gene, are necessary for the initial steps of bio-activation of PAHs (Rossner et al., 2020) to convert them into more redox-active products. It has also been shown that such mechanistic pathways differ substantially among different cells (Libalova et al., 2018; Vondráček et al., 2017). For example, it has been demonstrated that baseline esterase activities as well as secretion of cytochrome P450, which could markedly affect cellular metabolism, result in varied responses of murine and human cell lines to organic compounds (Veronesi and Ehrich, 1993). There is also a marked difference in the distribution of peroxisomal proteins (such as catalases) in human and mouse lung cells which could be responsible for different ROS activity in both types of cells (Karnati and Baumgart-Vogt, 2008). Therefore, a direct comparison between our results and those studies showing a significant role of PAHs in the oxidative stress is probably not reasonable.

Inorganic salts showed the lowest responses among all tested compounds and there was no significant difference in the responses ($p > 0.05$; one-way ANOVA) of any of these salts. Overall, at atmospherically relevant concentrations, inorganic salts seem to have very low contribution, if at all, to the oxidative stress as compared to the metals and quinones. This is consistent with previous studies based on ambient PM samples, showing either nil or inconsistent correlation of the macrophage ROS response with the concentration of inorganic ions (Hu et al., 2008; Kam et al., 2011; Verma et al., 2009; Wang et al., 2013; Xu et al., 2020).

## 4 Conclusion

In this paper, we have described the development of SCOPE for assessing the OP of water-soluble extracts of ambient PM in rat alveolar macrophages. The promising results of this instrument could pave the way for further development in automating other cellular assays. Moreover, since real-time instruments based on acellular OP assays have been developed in recent past, the current research opens up the road for the development of such online instruments based on mammalian cell lines, possibly coupling it to a real-time ambient PM sampling device (e.g., particle-into-liquid sampler or mist chamber). SCOPE is capable of analyzing up to 6 samples in a span of 5 hours without any manual intervention. The results of performance evaluation of the instrument demonstrate a high precision and accuracy for both positive control and the PM samples.

Overall, we have shown a first of its kind instrument capable of performing cellular OP measurements of PM. It substantially reduces the extent of manual labor associated with conducting cellular assays resulting in increased throughput of the results. We demonstrated that SCOPE is capable of handling large number of ambient PM samples, thus, providing an opportunity for generating an extensive dataset on cellular OP, that can be used in epidemiological studies. We also generated a database of several chemical compounds commonly known to be present in the ambient PM. Metals such as Fe (II), Mn (II) and Cu (II) dominated the OP, which were followed by quinones such as PQN and 1,2-NQN. PAHs and inorganic salts showed insignificant OP . Note, the ROS probe used in our study (DCFH-DA) does not measure the concentration of specific ROS (e.g., $H_2O_2$, $^{\bullet}OH$, $ROO^{\bullet}$, $O_2^{-\bullet}$, etc) separately, and therefore it is possible that despite a similar OP of the $PM_{2.5}$ chemical species as measured by SCOPE, the concentrations of the specific ROS, and the resulting health impacts caused by these ROS might be very different. Moreover, the reactivity of DCFH-DA to interact directly with the PM chemical components is not explored. Future studies should include specific measurement of different ROS using specific probes along with total OP to better understand the relationship between different chemical species and their health impacts.

*Data availability*. Supplementary data is provided with the manuscript.

*Author contributions*. SS developed the instrument, performed the experiments and prepared the manuscript. YW contributed in filters collection, helped in developing the manual protocol for cell-based experiments and edited the manuscript. JVP helped

in developing the instrument and edited the manuscript. VV conceived the idea, organized the manuscript and supervised the overall project.


*Competing interests*. The authors declare that they have no conflict of interest.

*Acknowledgments*. This work was supported by the National Science Foundation under Grant No. CBET-1847237. We thank Sandra McMasters, the director of cell media facility at UIUC for providing us NR8383 Cell Culture and media.

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

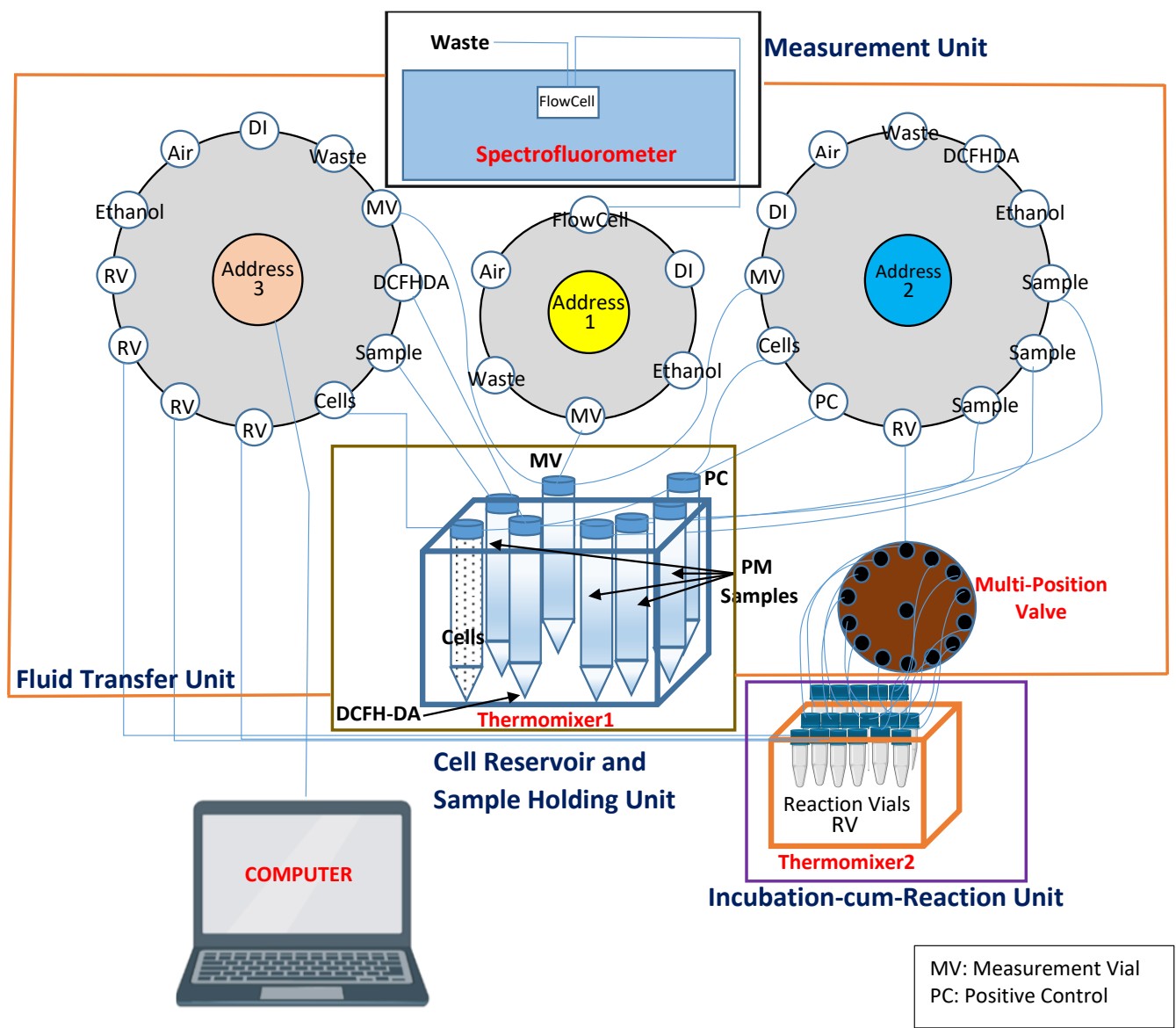

**Figure 1: Automated System Setup.** The instrument consists of four major units: cells reservoir and samples holder, fluid transfer unit, incubation-cum-reaction unit, and the measurement unit. The cells reservoir and sample holder unit consists of a set of several vials, containing cells, DCFH-DA solution, and the samples, all kept in Thermomixer 1. The fluid transfer unit consists of three syringe pumps (Pump #1, 2, and 3) and a 14-port multi-position valve connected to Pump #2. The incubation-cum-reaction unit consists of 17 Reaction Vials (RV), held in Thermomixer 2. The measurement unit consists of a spectrofluorometer equipped with a Flowcell.

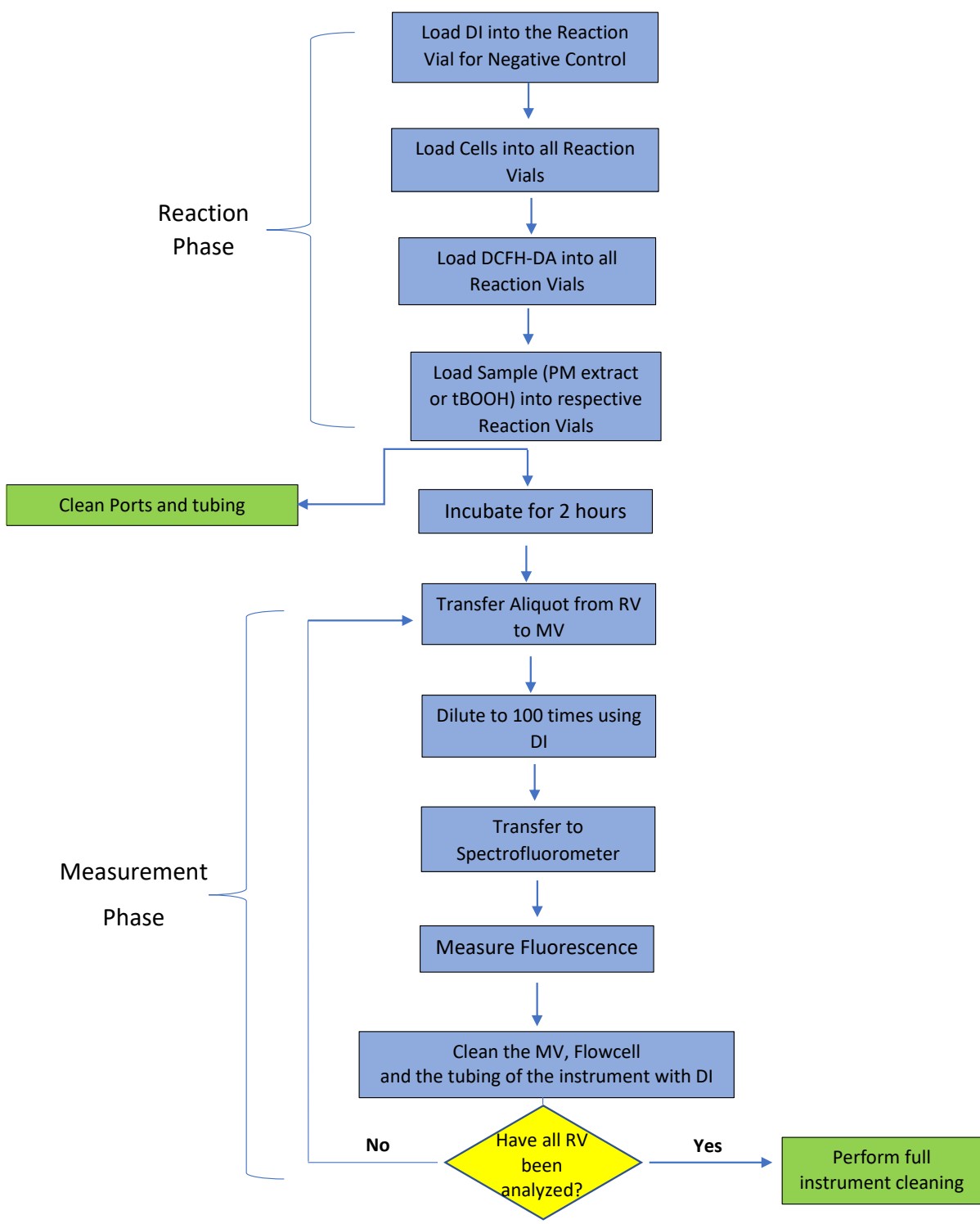

**Figure 2: Algorithm for the instrument's operational protocol.**

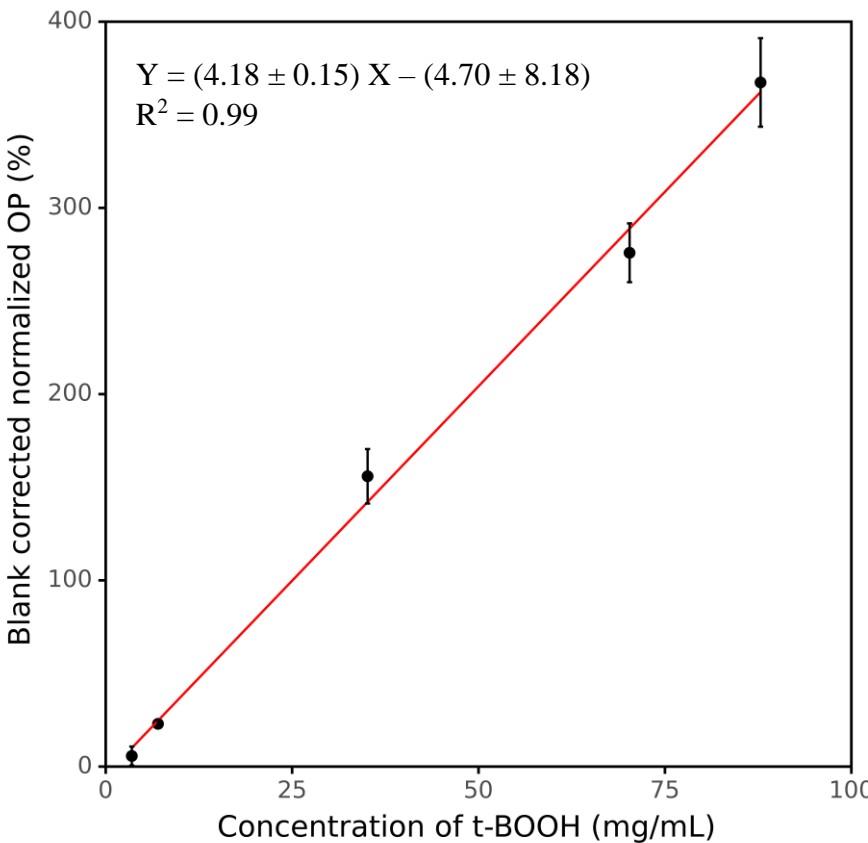

$$Y = (4.18 \pm 0.15)\, X - (4.70 \pm 8.18)$$
$$R^2 = 0.99$$


**Figure 3: OP as a function of the concentration of t-BOOH, measured by our automated instrument. The values on Y- axis were obtained by dividing absolute fluorescence of the sample by absolute fluorescence of negative control and then blank correcting it (i.e., subtracting 1 from ratio and then multiplying it by 100).**

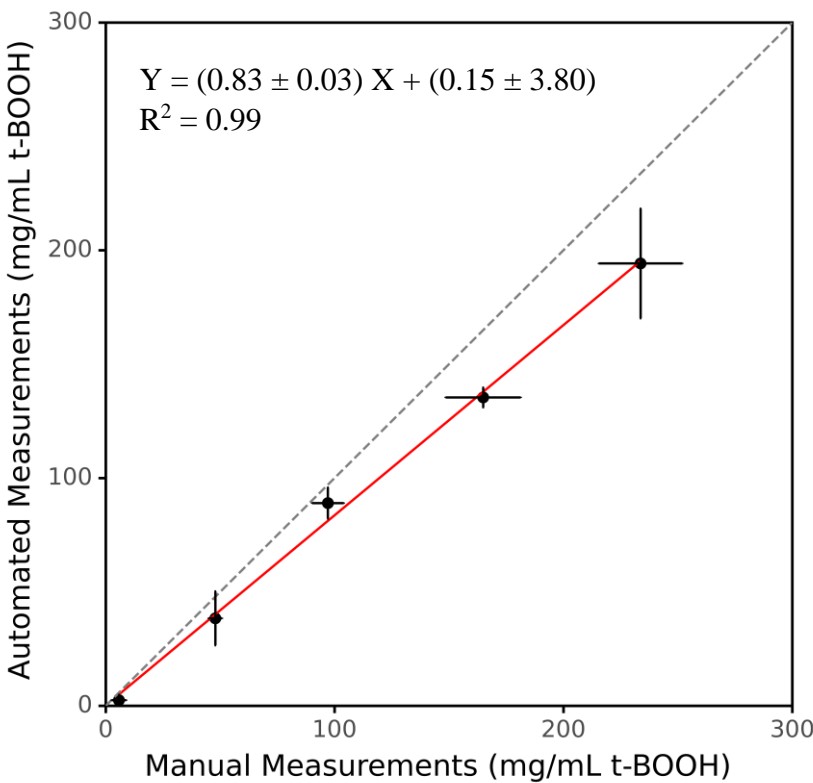

$$Y = (0.83 \pm 0.03) X + (0.15 \pm 3.80)$$
$$R^2 = 0.99$$

695 **Figure 4: Comparison between manual and automated measurements of OP for a positive control (t-BOOH). Dotted line represents the identity line.**

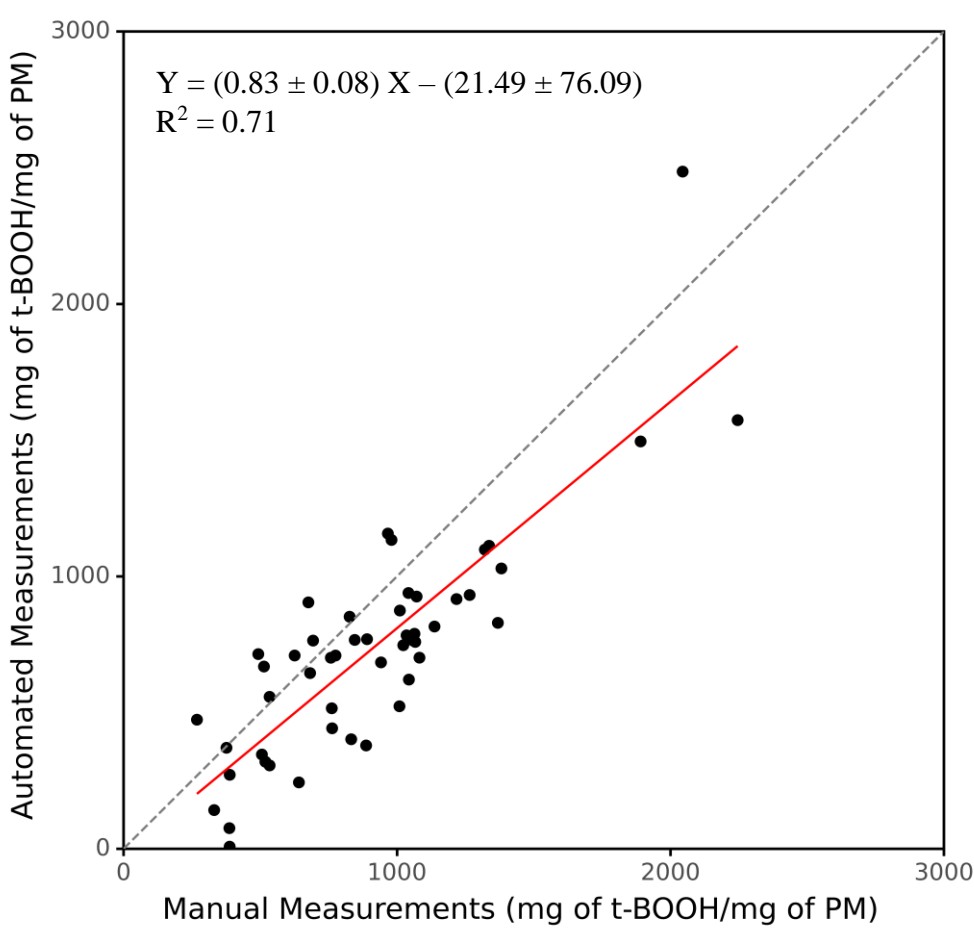

**Figure 5: Comparison of the OP for manual vs. automated operation using ambient PM samples (N=50). Dotted line represents the identity line.**



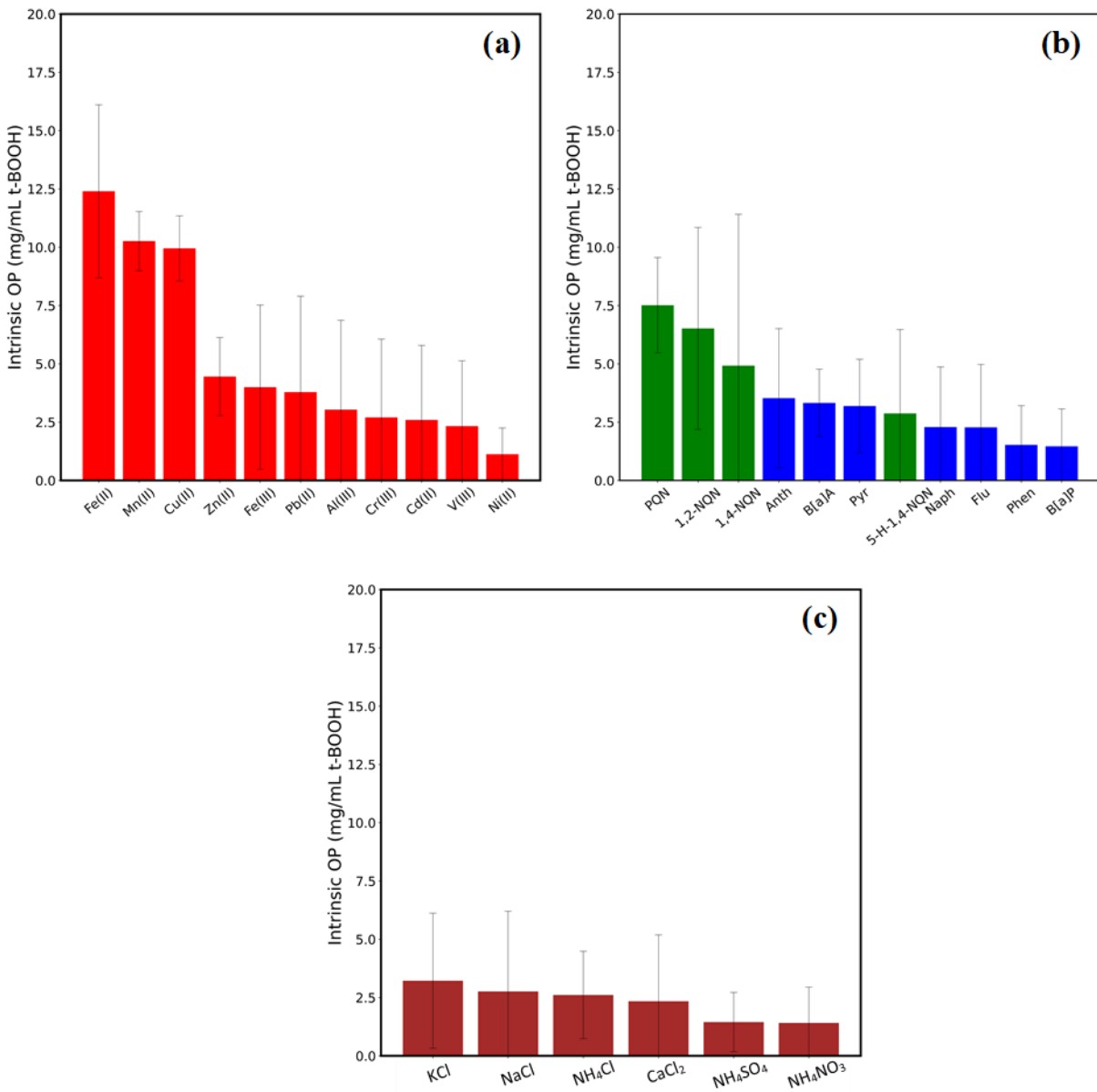


**Figure 6: Intrinsic OP of 11 transition metals (panel a), 4 quinones and 7 PAHs (panel b), and 6 inorganic salts (panel c). The concentration in the RV was 0.5 μM for all the metals; 0.2 μM for all quinones and PAHs, and 5 μM for inorganic salts, except for NaCl (1 μM). Error bars represent one standard deviation from the average.**


**Table 1: Limit of detection and precision of the instrument obtained through the measurements of field blanks, positive control and ambient PM samples (n = 10)**

| Sample | Unit | Average | Standard Deviation | LoD | CoV (%) |
|---|---|---|---|---|---|
| Field Blank | mg/mL t-BOOH | 5.30 | 0.42 | 1.26 | 7.95 |
| t-BOOH | % ROS response | 684.71 | 111.13 | - | 16.23 |
| Cu (II) | mg/mL t-BOOH | 71.05 | 10.18 | - | 14.33 |
| Zymosan | mg/mL t-BOOH | 18.84 | 7.15 | - | 37.97 |
| Ambient PM sample | mg of t-BOOH/mg of PM | 402.01 | 57.93 | - | 14.41 |
