# Peer review of "A Semi-automated Instrument for Cellular Oxidative Potential Evaluation (SCOPE) of Water-soluble Extracts of Ambient Particulate Matter"

_Atmospheric Measurement Techniques, 2021_

## Referee Comment (RC3)

Comments to "A Semi-automated Instrument for Cellular Oxidative Potential Evaluation (SCOPE) of Water-soluble Extracts of Ambient Particulate Matter" by Sudheer Salana et al. (MS No.: amt-2021-188).

This study introduced a semi-automated instrument for measuring cellular ROS formation potential (OP) of ambient PM and associated components in murine alveolar cells. This system was calibrated using dichlorofluorescein diacetate (DCFH-DA) as ROS probe and tert-Butyl hydroperoxide (t-BOOH) as standard compound for positive control. The authors found that metals, quinones, PAHs and inorganic salts exhibit different macrophage OP, claiming for the feasibility of using this system for assessing the cytotoxicity of different type of air pollutants. Overall the study is interesting and the topic fits the journal of AMT. However, the written of the manuscript needs some improvement before consideration of publishable potency. Detailed comments are as follows:

1. The authors need to justify and demonstrate why t-BOOH is chosen as standard compound for calibration.

2. Why choose rat alveolar macrophages? In previous studies, canine, human, and other different types of macrophages have been used as metrics (e.g. Beck-Speier et al., Oxidative stress and lipid mediators induced in alveolar macrophages by ultrafine particles. Free Radic. Biol. Med. 38, 1080-1092, 2005.). The calibrations in these studies were based on different standards. It is almost certain that OP of same PM samples from different macrophage assays will be different, including the current method. How do illustrate the baseline and OP differences across different methods?

3. The selectivity of the DCFH method toward different types of ROS should be discussed. If the ROS yields of certain concentrations of ambient PM and t-BOOH are the same, but the types of ROS (e.g. radicals and $H_2O_2$) formed by them are different, how to justify the health impact of ambient PM? The sensitivity/reactivity of the DCFH with different PM components (e.g. metal ions vs quinones) rather than with ROS should be considered and discussed.

4. Line 21 of page 1: Show the full name of PAH please. Whether oxygenated PAH is more accurate here? It looks like parent PAH generally do not exhibit prominent OP.

5. Line 41-43 of page 2: it is worthy to introduce the electron paramagnetic resonance (EPR) assay/method here.

6. Line 95-96: the 'one week' storing time is necessary? You may want to say use it up in one week or make fresh stocks each week.

7. Line 166 of page 6: Why '2 h incubation' is the best for measurement? In addition, for incubation of human macrophages, the mechanism and time period (much slower) for the metabolic process are quite different. More discussions are needed to clarify the gap between murine alveolar cells and human alveolar cells.

8. Line 181-192 of page 6-7: Clarify whether the filters have been prebaked (condition) or not?

9. Line 194 on page 7: The impact of sonication on ROS formation should be mentioned.

10. Line 197: What is the impact of fluorescent particle smaller than 0.45 μm in ambient particles to the measurement?

11. Line 235: the '1'' is confusing.

12. Sections 2.2 and 2.4 can be merged to form one section. Section 3.3 and 3.4 can be merged to form one section. The current Section 2.6 can be the last subsection in Section 2.

---

## Author Comment (AC1)

Manuscript ID: amt-2021-188                                            09/21/2021

Dr. Mingjin Tang

Associate Editor Atmospheric Measurement Techniques

Dear Dr. Mingjin Tang,

Along with this letter, we have submitted our response document for the manuscript "A Semi-automated Instrument for Cellular Oxidative Potential Evaluation (SCOPE) of Water-soluble Extracts of Ambient Particulate Matter". All the comments raised by the reviewers have been satisfactorily addressed based on a point-by-point response in the attached document. Additional experiments are performed to address some of the comments and their results have been included in the manuscript. To facilitate the review process, we have also included the marked-up version of our revised manuscript (track-changes mode), so that the reviewers can see how the comments are incorporated in the manuscript. The manuscript has been substantially improved as a result of this review and we really appreciate all the valuable suggestions provided by the reviewers.

We believe that our revised manuscript meets the high-quality standards of AMT, and we look forward to any further comments the reviewers and editor might have.

Sincerely,

Sudheer Salana

Graduate Student
Department of Civil and Environmental Engineering
University of Illinois at Urbana-Champaign

N Mathews Ave, Urbana, IL 61801

Reviewer #1: Salana et al. work presented an automated syringe-pump system for assessing the ROS generation from alveolar macrophage when incubated with different samples. The manuscript has discussed the setup, running procedures, LOD, precision, comparison to manual method, and the calibration of the system. I think this is a very unique study that can be inspiring to many readers on AMT. I recommend acceptance after the authors address the following minor comments.

1.  A batch of samples can take up to 5 hours as mentioned in the manuscript. This means the cell suspensions are left in the system for up to 5 hours. How healthy cells after sitting in an environment outside of the incubator for a couple hours? Do cell numbers change over time? The authors should add some discussions regarding this.

Response
We thank the reviewer for this suggestion. Before, designing the protocol of our instrument, we conducted an experiment by keeping the cells outside an incubator but in a temperature-controlled environment (i.e., 37 °C maintained through a thermomixer used in our instrument), and measured the cell viability using trypan blue. The results of this experiment are shown in Fig. R1. We found that over a period of 5 hours, the cell viability decreased by only 6%. However, the cell viability starts decreasing sharply beyond 6 hours. Therefore, we limited the cells exposure to the outside environment for only five hours. In fact, the results of this experiment were the basis for limiting the maximum number of the samples (N=6), that can be analyzed in one batch of this instrument. Based on the reviewer's suggestion, we have added figure S1 (and the details of this experiment) in the supplementary information of the revised manuscript, showing the variations in cell viability as a function of time, and added this discussion on Page 4, lines 121-125 of the revised manuscript. *"Before designing the protocol of our instrument, we conducted an experiment by keeping the cells outside an incubator but in a temperature-controlled environment (i.e., 37 °C maintained through a thermomixer used in our instrument) and measured the cell viability using trypan blue [see Figure S1 in the supplementary information (SI)]. We found that over a period of 5 hours, the cell viability decreased by only 6%. However, the cell viability started decreasing sharply beyond 6 hours. Therefore, we limited the cells exposure to the outside environment for only five hours."*

[Figure]

**Figure R1:** Variation in cell viability [(viable cells/total cells) *100] for NR8383 cells suspended in 1XSGM at 37 ℃ (outside an incubator) as a function of time. Cell Viability was measured using Trypan Blue Assay. 100 µL of 0.4% trypan blue solution was mixed with 100 µL of cells and incubated for 3 minutes at room temperature. After incubation, 10 µL of the mixture was withdrawn, applied to a hemocytometer and viable cells (unstained cells) were counted under a microscope. Error bars denote one standard deviation of the average (N=3).

2.      Line 227, fig 4 should be fig 3.

Response
We apologize for this mistake. On Line 261 of the revised manuscript, Fig. 4 has been changed to Fig. 3.

3.      line 231, I agree with what the authors say about express LOD in terms of standards but providing a rough liquid concentrations or doses of PM extracts can be very helpful to readers. This gives ideas of how much mass is required to have a signal above detection limit.

Response
Based on our extensive measurements, we find it generally hard to detect a signal if the PM concentration in our liquid extract is less than 20 µg/mL. Therefore, we have added the following sentences to the manuscript on Page 9, Line 265-267.

*"Nevertheless, based on several experiments, we found that it is difficult to detect a signal for a PM extract with concentration below 20 µg/mL, which could be considered as a rough detection limit for SCOPE."*

4.      line 255 "0.04 to 9.75 mM" please use mg/mL to keep consistency in units.

Response
All the units have been converted to mg/mL

5.      Fig 6, error bars seem quite high. Please provide statistical analysis.

Response
Based on the reviewer's suggestion, we have conducted a one-way ANOVA (analysis of variance) test followed by Tukey's test for post-hoc analysis on the intrinsic OP responses of different groups of the species, i.e., the metals, organic compounds and inorganic compounds. Based on these results, we have added following sentences on Page 11, Lines 323-336, Lines 358-359, Lines 351-353 and Page 12, Lines 371-372 of the revised manuscript:

*"To assess significant differences in the OP responses, we used a one-way ANOVA (analysis of variance) test followed by Tukey's test for post-hoc analysis on the intrinsic OP responses of different groups of the species, i.e., metals, organic and inorganic compounds. Among metals, Fe (II), Mn (II), and Cu (II) induced the highest response (12.40 -9.95 mg/mL t-BOOH). Although, the OP of these three metals were not statistically different from each other, their responses were significantly different from the rest of the metals (p<0.05). Other metals [Fe (III), Zn (II), Pb (II), Al (III), Cr (III), Cd (II) and V(III)] induced very low response (<4.5 mg/mL t-BOOH), and there was no statistical difference among their responses (p>0.05). Interestingly, the pattern of Fe (III) vs. Fe (II) OP response (~3 times lower response of Fe III than Fe II) matches with their relative redox activities as measured by the dithiothreitol (DTT) assay, i.e., 3 times lower intrinsic DTT activity of Fe (III) compared to Fe (II) (Charrier and Anastasio, 2012).*

*Among the organic compounds, PQN and 1,2-NQN showed the highest response (7.51 and 6.52 mg/mL t-BOOH, respectively), however, their responses were significantly lower (p>0.05) than that of the metals Fe (II), Mn (II) and Cu (II). Other than these two quinones, the OP of any of the organic compounds, i.e PAHs, 1, 4-NQN and 5-H-1,4-NQN was not significantly above the negative control."*

*"Inorganic salts showed the lowest responses among all tested compounds and there was no significant difference in the responses (p>0.05; one-way ANOVA) of any of these salts. "*

Reviewer # 2: The authors present a new semi-automated instrument to assess cellular oxidative potential (OP) when exposed to particulate matter, based on the DCFH-DA assay, which is capable of analyzing six samples in only 5 hours. Furthermore, they investigate the intrinsic OP of a range of standards which are of interest with respect to ambient PM OP. The authors discuss the functionality of the method, as well as the operational procedure, calibration, limit of detection and reproducibility. This is a novel and interesting method for quantifying cellular OP representing a significant technical advancement, and certainly fits the scope of AMT. I recommend publication after considering the following minor comments:

Line 122 – It is unclear what the negative control actually is, please elaborate

Response
The negative control was always the deionized Milli-Q water (DI). We have clarified this on Page 5, Line 153.

Line 161 – Why specifically was tertbutyl hydroperoxide chosen as the positive control as opposed to e.g. H2O2?

Response
Tertbutyl hydroperoxide (t-BOOH) is a well-established inducer of the cellular oxidative stress. t-BOOH diffuses through the cell membrane quite efficiently and has been demonstrated to induce a comprehensive oxidative stress response through the generation of a variety of species including $H_2O_2$, alkoxyl and peroxyl radicals. For example, t-BOOH is more stable in cellular systems compared to $H_2O_2$ which can easily undergo degradation by catalases (cellular enzymes that protect cells from oxidative damage) and therefore is a better positive control to understand cellular defense mechanisms (Abe and Saito 1998). t-BOOH has also been found to be a better at glutathione (GSH) depletion as compared to other oxidants (Dierickx et al., 1999), inhibiting peroxiredoxin (an antioxidant protein that protects certain enzymes from oxidative damage) activity (Ikeda et al., 2011), evoke a more consistent cellular antioxidant response (Alia et al., 2005), cause a greater DNA damage than $H_2O_2$ (Slamenova et al., 2013) and promote a more efficient peroxidation of membrane lipids as compared to $H_2O_2$ (Guidarelli et al., 1997). t-BOOH is also a better model for the organic hydroperoxides that are formed when the cellular fatty acids and proteins react with oxygen during pathological conditions (Chance et al., 1979). All these properties of t-BOOH make it an excellent positive control. There are other positive controls such as Menadion, which are used in pharmacological studies, however the low cost and easy availability of t-BOOH makes it a better choice.

The following lines have been added to manuscript on Page 10, Line 284-292:

*"t-BOOH is a well-established inducer of oxidative stress, not only in macrophages (Lopes et al., 2017; Prasad et al., 2007; Roux et al., 2019) but also in a variety of other cells such as hepatocytes (Kučera et al., 2014), sperm cells (Fatemi et al., 2012), and lung fibroblast cells (Lopes et al., 2017). t-BOOH diffuses through the cell membrane quite efficiently and has been demonstrated to induce a comprehensive oxidative stress response through the generation of a variety of species including $H_2O_2$, alkoxyl and peroxyl radicals. t-BOOH has also been found to be more stable in*

*the cellular systems (Abe and Saito 1998), and also a better at glutathione (GSH) depletion (Dierickx et al., 1999), inhibiting peroxiredoxin activity (Ikeda et al., 2011), evoke a more consistent cellular antioxidant response (Alia et al., 2005), cause a greater DNA damage (Slamenova et al., 2013) and promote a more efficient peroxidation of membrane lipids as compared to other oxidants such as $H_2O_2$ (Guidarelli et al., 1997)."*

Line 166 – Is a DCFH-DA control performed alongside each 2 hour cell measurement, or before the batch 6 batches of cells are analysed? Is there any change in the DCFH-DA stock reactivity over the 5-hour period that could complicate quantification due to degradation etc?

Response
Previous studies have indicated that DCFH-DA is generally a stable probe for at least a period of 2-3 hours (Landreman et al., 2008). Moreover, it has been shown that DCFH-DA is highly stable in HEPES buffer [used in our Salt Glucose Media (SGM)] and does not show any autooxidation in such culture media (Le Bel and Bondy, 1990; Arbogast and Reid, 2004). Therefore, we did not perform a DCFH-DA control alongside the 2-hour ROS measurement. However, to further confirm these findings and to address the reviewer's comment, we conducted an experiment in our lab to measure the variations in absolute fluorescence of DCFH-DA as a function of time to assess its degradation or autooxidation. In this experiment DCFH-DA was prepared as discussed in Section 2.2 of the manuscript and transferred to two different amber vials. One of these vials was stored in the thermomixer at 37 °C and the other vial was stored at room temperature (23 °C). Changes in fluorescence of DCFH-DA in each vial was measured at every 30 minutes, for a period of up to 6 hours. The results of this experiment are shown in Fig. R2.

[Figure]

**Figure R2:** Variation in the absolute fluorescence of DCFH-DA as a function of time. DCFH-DA Error bars denote one standard deviation of the mean (N=3 replicates).

As can be seen in Fig. R2 the absolute fluorescence of DCFH-DA remains almost constant in either condition which indicates that there is no appreciable degradation of DCFH-DA within 5-hour period. We have added this figure in the SI (Fig. S2) and the related discussion on Page 5, Line 131-135:

*"We measured the variation in absolute fluorescence of DCFH-DA as a function of time to assess the possible degradation or autooxidation of DCFH-DA during our measurement. The results showed that the absolute fluorescence of DCFH-DA remains constant for a period of at least 6 hours, indicating the stability of the probe within our experimental timeframe (please refer to Fig. S2 in SI)."*

Line 227 – should this be Figure 3?

Response
Yes, we apologize for our mistake. This is Fig. 3. On Line 261 of the revised manuscript, Fig. 4 has been changed to Fig. 3.

Line 234 – mg/ml and µM units are used interchangeable through the manuscript, consistent units would be beneficial for comparison.

Response
All the units have been changed to mg/mL for consistency.

Line 237 – what values were used for PM normalization, the extracted PM mass in mg/ml?

Response
We apologize for the confusion. The ROS response for the PM samples was normalized by *concentration of the PM extract* (and not by the PM mass) in the RV. Since, final concentration of the PM in RV for the precision experiment was 30 µg/ mL, we normalized the ROS response by this value, i.e., 0.03 mg/mL, to obtain the final results in the units of mg of t-BOOH per mg of PM. We have provided this detail in the SI of the manuscript (section S1). We have also corrected it in the manuscript, on Page 9, lines 273.

Figure 1 – This Figure could benefit from a more descriptive Figure caption to make it easier to follow the schematic.

Response
The following paragraph has been added to the caption.
*"The instrument consists of four major units: cells reservoir and samples holder, fluid transfer unit, incubation-cum-reaction unit, and the measurement unit. The cells reservoir and sample holder unit consists of a set of several vials, containing cells, DCFH-DA solution, and the samples, all kept in Thermomixer 1. The fluid transfer unit consists of three syringe pumps (Pump #1, 2, and 3) and a 14-port multi-position valve connected to Pump #2. The incubation-cum-reaction unit consists of 17 Reaction Vials (RV), held in Thermomixer 2. The measurement unit consists of a spectrofluorometer equipped with a Flowcell."*

Figure 6 – The three panels in the Figure should be labelled A-C.

Response
We thank the reviewers for this suggestion. The panels have been labelled as a-c.

Figure 6 – The error bars associated with Figures 6 A-C are in some cases quite large, could the authors comment on the source of this variability?

Response
The error bars are mostly high for the species, which have intrinsic OP less than 5 mg/mL t-BOOH. This is probably due to low sensitivity of the instrument at that range which causes an amplification of variability when the OP response is closer to the detection limit. We could have tried to increase the concentration of these species to reliably measure their intrinsic OP, however, that would make these concentrations beyond the typical range for their atmospherically relevant levels. Essentially, the low intrinsic activity with high error bars indicates a very low contribution of these species in the overall cellular OP measured by the macrophage ROS assay, at their atmospherically relevant concentrations.

Reviewer # 3 - This study introduced a semi-automated instrument for measuring cellular ROS formation potential (OP) of ambient PM and associated components in murine alveolar cells. This system was calibrated using dichlorofluorescein diacetate (DCFH-DA) as ROS probe and tert-Butyl hydroperoxide (t-BOOH) as standard compound for positive control. The authors found that metals, quinones, PAHs and inorganic salts exhibit different macrophage OP, claiming for the feasibility of using this system for assessing the cytotoxicity of different type of air pollutants. Overall the study is interesting and the topic fits the journal of AMT. However, the written of the manuscript needs some improvement before consideration of publishable potency. Detailed comments are as follows:

1. The authors need to justify and demonstrate why t-BOOH is chosen as standard compound for calibration.

Response
This comment is similar to the comment # 2 raised by the $2^{nd}$ reviewer, therefore, we are reproducing our response here again.
"Tertbutyl hydroperoxide (t-BOOH) is a well-established inducer of oxidative stress. t-BOOH diffuses through the cell membrane quite efficiently and has been demonstrated to induce a comprehensive oxidative stress response through the generation of a variety of species including $H_2O_2$, alkoxyl and peroxyl radicals. For example, t-BOOH is more stable in cellular systems compared to $H_2O_2$ which can easily undergo degradation by catalases (cellular enzymes that protect cells from oxidative damage) and therefore is a better positive control to understand cellular defense mechanisms (Abe and Saito 1998). t-BOOH has also been found to be a better at glutathione (GSH) depletion as compared to other oxidants (Dierickx et al., 1999), inhibiting peroxiredoxin (an important antioxidant protein that protects certain enzymes from oxidative damage) activity (Ikeda et al., 2011), evoke a more consistent cellular antioxidant response (Alia et al., 2005), cause a greater DNA damage than $H_2O_2$, (Slamenova et al., 2013) and promote a more efficient peroxidation of membrane lipids as compared to $H_2O_2$, (Guidarelli et al., 1997). t-

BOOH is also a better model for the organic hydroperoxides that are formed when the cellular fatty acids and proteins react with oxygen during pathological conditions (Chance et al., 1979). All these properties of t-BOOH make it an excellent positive control. There are other positive controls such as Menadion, which are used in pharmacological studies, however the low cost and easy availability of t-BOOH makes it a better choice."

We have added this discussion in our manuscript (Page 10, Line 284-292).

2. Why choose rat alveolar macrophages? In previous studies, canine, human, and other different types of macrophages have been used as metrics (e.g. Beck-Speier et al., Oxidative stress and lipid mediators induced in alveolar macrophages by ultrafine particles. Free Radic. Biol. Med. 38, 1080-1092, 2005.). The calibrations in these studies were based on different standards. It is almost certain that OP of same PM samples from different macrophage assays will be different, including the current method. How do illustrate the baseline and OP differences across different methods?

Response
We agree with the reviewer that OP analysis of the same PM samples from different macrophage assays will yield different results. A number of previous studies have indeed used macrophages of canine, human, hamster and murine origin. However, rat macrophages (particularly NR8383) are still one of the most widely used cell lines in the PM studies and therefore, its use in our instrument makes it easier for comparison among different studies. Certain characteristics of this cell line make it one of the best macrophage models available for the evaluation of OP. These characteristics include minimal maintenance (can be studied in a BSL-1 lab) and highly reproducible results that are comparable to primary cells (Helmke et al., 1988). Moreover, NR8383 is superior for studying inflammatory responses and immune defense system compared to commonly used cell lines such RAW264.7 (murine), A549, U937 and THP-1 (all human macrophage cell lines). This is because unlike other cell lines, it has the ability to express the Mannose Receptor, which is a key protein linked to macrophage function (Lane et al., 1998). NR8383 also expresses a number of inflammatory cytokines such as IL-1β and TNF-α (Lin et al., 2000), thus it will allow us to link the results obtained from this instrument to these inflammatory responses, in our future studies.

As the reviewer has pointed out, establishment of a baseline and comparison of OP across different cell lines is a difficult task. This will require a systematic comparison of different cell lines with different types of PM samples, and as such will be a huge analysis effort by itself. Our automated instrument is a small but an important step in the direction of facilitating such measurements. At present, the instrument uses rat alveolar macrophages, however, in the future, we can possibly customize it to use for other cell lines as well. This will really help in making a systematic comparison among different cell lines and hopefully establishing a baseline. However, it is beyond the scope of our current study.

Considering the reviewer's suggestion, we have added following sentences in the revised manuscript on Page 4, Line 109-114:

*"We have used a murine cell line, NR8383, as it is one of the most widely used cell lines in the PM studies. Certain characteristics of this cell line make it one of the best macrophage models available for the evaluation of PM OP. These characteristics include minimal maintenance (can be studied in a BSL-1 lab) and highly reproducible results that are comparable to primary cells (Helmke et al., 1988). NR8383 also expresses a number of inflammatory cytokines such as IL-1β and TNF-α (Lin et al., 2000), thus it will allow us to link the results obtained from this instrument to these inflammatory responses, in our future studies"*

3. The selectivity of the DCFH method toward different types of ROS should be discussed. If the ROS yields of certain concentrations of ambient PM and t-BOOH are the same, but the types of ROS (e.g. radicals and H2O2) formed by them are different, how to justify the health impact of ambient PM? The sensitivity/reactivity of the DCFH with different PM components (e.g. metal ions vs quinones) rather than with ROS should be considered and discussed.

Response
DCFH-DA is a non-specific ROS probe. Although it was originally believed that DCFH-DA was specific to $H_2O_2$ (Keston and Brandt, 1965), this was not the case as found in a later study (Le Bel et al., 1992). Since a broad range of oxygen species oxidize DCFH, it provides a general assessment of the overall redox state of the cells rather than a quantitative estimate of the specific ROS. We agree with the reviewer on the conundrum posed by measurement of total ROS. Indeed, it is possible that even though the total ROS of two different PM samples is the same, but the concentrations of specific ROS, and the resulting health impacts caused by these ROS might be very different. This is a valid concern about the use of such comprehensive ROS probes, but we don't think that we can answer this question based on our study. This will require a simultaneous measurement of different ROS using different probes and their systematic comparison with either the toxicological or epidemiological endpoints, to understand the relative importance of these different ROS.

We also agree with the reviewer that DCFH-DA might be more sensitive to certain chemical species than others, which could influence the intrinsic OP results shown in Figure 6. However, the main focus of our present study is to develop an automated instrument which can imitate a well-established manual protocol for the cellular ROS measurement and demonstrate its application by measuring the intrinsic OP of various PM chemical species that can interact with the macrophages to generate ROS. Evaluating the nature and preferences of DCFH-DA to directly react with the chemical species is beyond the scope of this paper as that would require a more thorough investigation of the numerous molecular pathways of both deacetylation of DCFH-DA as well as the oxidation of DCFH (Burkitt and Wardman, 2001; Bonini et al., 2006, Hempel et al., 1999). Without such evaluation, we fear, any discussion on the specificity of DCFH-DA to chemical species will be speculative. However, we do intend to explore these relationships between DCFH-DA and PM chemical species in the near future. Nevertheless, based on the reviewer's suggestions, we have included the following brief discussion along these points in our manuscript on page 12, line 379-384:

*"Note, the ROS probe used in our study (DCFH-DA) does not measure the concentration of specific ROS (e.g., $H_2O_2$, $OH^{\bullet}$, $ROO^{\bullet}$, $^{\bullet}O_2^{-}$, etc.) separately, and therefore it is possible that despite*

*a similar OP of the PM₂.₅ chemical species as measured by SCOPE, the concentrations of the*
*specific ROS, and the resulting health impacts caused by these ROS might be very different.*
*Moreover, the reactivity of DCFH-DA to interact directly with the PM chemical components is not*
*explored. Future studies should include specific measurement of different ROS using specific*
*probes along with total OP to better understand the relationship between different chemical*
*species and their health impacts."*

4. Line 21 of page 1: Show the full name of PAH please. Whether oxygenated PAH is more
accurate here? It looks like parent PAH generally do not exhibit prominent OP.

Response
Full name of PAH has been added to Line 22. We agree with the reviewer that oxygenated products
of the PAHs could be more OP-active than the parent PAHs, as also indicated in some of the
studies (Gurbani et al., 2013; Sklorz et al., 2007; Wang et al., 2011). However, our focus here was
to evaluate some of the most common and priority PAHs as defined by USEPA (Husar et al.,
2012), which are known to be present in the ambient PM. We intend to explore more PAHs and
the effect of oxidation in a more systematic way (e.g., in a oxidation flow reactor) in the future.

5. Line 41-43 of page 2: it is worthy to introduce the electron paramagnetic resonance (EPR)
assay/method here.

Response
We thank the reviewer for this suggestion. We have added EPR assay in Line 44-45 on Page 2.
*"and electro paramagnetic resonance (EPR) measurements (Dikalov et al., 2018; Jeong et al.,*
*2016)".*

6. Line 95-96: the 'one week' storing time is necessary? You may want to say use it up in one
week or make fresh stocks each week.

Response
No, one week of storing time is not necessary. The structure of our sentence was not clear here.
This sentence has been changed to *"The stock solutions of quinones (PQN, 1,2-NQN, 1,4-NQN)*
*were prepared in DMSO, stored in a freezer at -20 ℃ and used within a week."* On Page 4 Line
of the revised manuscript.

7. Line 166 of page 6: Why '2 h incubation' is the best for measurement? In addition, for
incubation of human macrophages, the mechanism and time period (much slower) for the
metabolic processes are quite different. More discussions are needed to clarify the gap between
murine alveolar cells and human alveolar cells.

Response
Before we determined the protocol for our automated instrument, we tested the kinetics of ROS
generation for two randomly chosen PM samples from the sample set analyzed in our study, by
measuring the ROS response at every half an hour till 3.5 hours. The results of this analysis have
been included in the supplemental information (Fig. S3) of the revised manuscript and are reproduced here (Fig. R3). As can be seen, the ROS response peaks and stabilizes at around 2-hour incubation time for both of the PM samples. Note, these results are consistent with Landreman et al., 2008, which also reported that for most samples (PM, blanks, positive control), the ROS response stabilize at around 2-hour incubation time. Therefore, we chose 2 hours of incubation time for our measurement.

[Figure]

**Figure R3:** Effect of incubation time on the OP of PM samples. Each measurement was performed in triplicates. Error bars denote one standard deviation of the mean.

We have also added the following text in the revised manuscript on Page 5, Line 138-143:

"*The incubation time of 2 hours was chosen after measuring the kinetics of ROS generation for two PM samples (chosen randomly from the sample set analyzed in our study) at a time interval of 30 minutes over a 3.5 h time period (please refer to Fig. S3 in SI). It was found that the ROS response peaks and stabilizes at around 2-hour incubation time for both of the PM samples. These results are consistent with Landreman et al., (2008), which also reported that for most samples (PM, blanks, positive control), the ROS response stabilizes at around 2-hour incubation time.*"

We agree with the reviewer that metabolic processes in human cells could be quite different from those in murine cells and this could also be one of the reasons why PAHs showed much lower OP in our study. We have added the following sentences in the revised manuscript to clarify the gap between murine and human cells on Page 11, Line 350-354:

*"For example, it has been demonstrated that baseline esterase activities as well as secretion of cytochrome P450, which could markedly affect cellular metabolism, result in varied responses of murine and human cell lines to organic compounds (Veronesi and Ehrich, 1993). There is also a marked difference in the distribution of peroxisomal proteins (such as catalases) in human and mouse lung cells, which could be responsible for different ROS activity in both types of cells (Karnati and Baumgart-Vogt, 2008).*

8. Line 181-192 of page 6-7: Clarify whether the filters have been prebaked (condition) or not?

Response
All the filters were prebaked at 550 °C.
The following sentences has been added in the revised manuscript (Page 6, Line 174): *"All the filters were prebaked at 550 °C for 24 hours before sampling."*

9. Line 194 on page 7: The impact of sonication on ROS formation should be mentioned.

Response
In our analysis, we found that ROS response of a blank filter extracted in DI by sonication was only slightly higher than that of DI (average ratio of blank filter to DI = 1.17± 0.02; N= 20). Moreover, we always blank corrected the ROS response of a PM sample with that of the field blank filter. Therefore, any effect of sonication caused by the extraction of filter in DI should have been largely cancelled out. We have added following sentences in the revised manuscript on Page6, Line 185-198:

*"Although sonication could potentially lead to the formation of ROS (Miljevic et al., 2014), we found that ROS response of a blank filter extracted in DI by sonication was only slightly higher than that of DI (average ratio of blank filter to DI = 1.17± 0.02; N= 20). Moreover, we always blank corrected the ROS response of a PM sample with that of the field blank filter. Therefore, any effect of sonication caused by the extraction of filter in water should have been largely cancelled out."*

10. Line 197: What is the impact of fluorescent particle smaller than 0.45 μm in ambient particles to the measurement?

Response
This is a valid comment. Following the reviewer's point, we conducted the experiments to quantify the impact of fluorescent particle smaller than 0.45 μm in the ambient PM. Specifically, we extracted 10 randomly chosen PM samples from the sample set analyzed in our study, extracted them in DI, filtered the extracts through a 0.45 μm syringe filter, and measured their fluorescence at the same wavelengths (excitation 488 nm/ emission 530 nm) as used for the DCF measurement. The difference between absolute fluorescence of the filtered extracts ($0.52 \pm 0.04$ fluorescence units) and DI ($0.47 \pm 0.1$ fluorescence units) was not statistically significant (p> 0.05; unpaired t-test). Moreover, absolute fluorescence of the filtered PM extract was 60-80 times lower than that of a negative control (i.e., DI+cells+DCFH-DA). Therefore, we conclude that contribution of the fluorescent ambient particles smaller than 0.45 μm to the ROS measurement is negligible.

We have also added following text in the revised manuscript on Page 7, Line 189-195:

*"We also assessed the impact of fluorescent particle smaller than 0.45 µm in our ambient PM extracts. Specifically, we extracted 10 randomly chosen PM samples from the sample set analyzed in our study, extracted them in DI, filtered the extracts through a 0.45 µm syringe filter, and measured their fluorescence at the same wavelengths (excitation 488 nm/ emission 530 nm) as used for DCF. The difference between absolute fluorescence of the filtered extracts ($0.52 \pm 0.04$ fluorescence units) and DI ($0.47 \pm 0.1$ fluorescence units) was not statistically significant ($p > 0.05$; unpaired t-test). The absolute fluorescence of the filtered PM extract was 60-80 times lower than that of a negative control. Thus, the contribution of fluorescent ambient particles smaller than 0.45 µm to the ROS measurement is negligible."*

11. Line 235: the '1'" is confusing.

Response
*1"* has been replaced with *1 inch. (Page 9, Line 271).*

12. Sections 2.2 and 2.4 can be merged to form one section. Section 3.3 and 3.4 can be merged to form one section. The current Section 2.6 can be the last subsection in Section 2.

Response
We thank the reviewer for this suggestion. We have merged these sections. We have also made the current Section 2.6 as the last subsection of Section 2 (Section 2.7).

**References cited in the response document**

Abe, K., and Saito, H: Characterization of t-butyl hydroperoxide toxicity in cultured rat cortical neurones and astrocytes, Pharmacology and Toxicology, *83*(1), 40–46, https://doi.org/10.1111/j.1600-0773.1998.tb01440.x, 1998.

Alía, M., Ramos, S., Mateos, R., Bravo, L. and Goya, L.: Response of the antioxidant defense system to tert-butyl hydroperoxide and hydrogen peroxide in a human hepatoma cell line (HepG2), J. Biochem. Mol. Toxicol., *19*(2), 119–128, https://doi.org/10.1002/jbt.20061, 2005.

Arbogast, S., and Reid, M. B.: Oxidant activity in skeletal muscle fibers is influenced by temperature, CO2 level, and muscle-derived nitric oxide, Am. J. Physiol. Regul. Integr. Comp. Physiol., 287(4), 698-705, https://doi.org/10.1152/ajpregu.00072.2004, 2004.

Bonini MG, Rota C, Tomasi A and Mason RP.: The oxidation of 2′,7′-dichlorofluorescein to reactive oxygen species: a self-fulfilling prophesy? Free Radic. Biol. Med., 40(6), 968–975, https://doi.org/10.1016/j.freeradbiomed.2005.10.042, 2006.

Burkitt MJ and Wardman P.: Cytochrome c is a potent catalyst of dichlorofluorescein oxidation: implications for the role of reactive oxygen species in apoptosis, Biochem. Biophys. Res. Commun., 282 (1), 329–333. https://doi.org/10.1006/bbrc.2001.4578, 2001.

Chance, B., Sies, H., and Boveris, A.: Hydroperoxide metabolism in mammalian organ, Physiol. Rev., *59*(3), 527–605, https://doi.org/10.1152/physrev.1979.59.3.527, 1979.

Charrier, J. G., and Anastasio, C. (2012).: On dithiothreitol (DTT) as a measure of oxidative potential for ambient particles: Evidence for the importance of soluble \newline transition metals, Atmos. Chem. Phys., 12(19), 9321–9333, https://doi.org/10.5194/acp-12-9321-2012, 2012.Dierickx, P. J., Van Nuffel, G., and Alvarez, I.: Glutathione protection against hydrogen peroxide, tert-butyl hydroperoxide and diamide cytotoxicity in rat hepatoma-derived Fa32 cells, Hum. Exp. Toxicol., *18*(10), 627–633, https://doi.org/10.1191/096032799678839482,1999.

Dikalov, S. I., Polienko, Y. F., & Kirilyuk, I.: Electron Paramagnetic Resonance Measurements of Reactive Oxygen Species by Cyclic Hydroxylamine Spin Probes, Antioxid. Redox Signal., *28*(15), 1433–1443, https://doi.org/10.1089/ars.2017.7396, 2018.

Guidarelli, A., Cattabeni, F., and Cantoni, O.: Alternative mechanisms for hydroperoxide-induced DNA single strand breakage, Free Radic. Res., 26(6), 537-547, https://doi.org/10.3109/10715769709097825, 1997.

Gurbani, D., Bharti, S. K., Kumar, A., Pandey, A. K., Ana, G. R., Verma, A., Khan, A.H., Patel, D.K., Mudiam, M.K.R., Jain, S.K., Roy, R., and Dhawan, A.: Polycyclic aromatic hydrocarbons and their quinones modulate the metabolic profile and induce DNA damage in human alveolar and bronchiolar cells, Int. J. Hyg. Environ. Health, 216(5), 553-565, https://doi.org/10.1016/j.ijheh.2013.04.001, 2013.

Hempel SL, Buettner GR, O'Malley YQ, Wessels DA and Flaherty DM.: Dihydrofluorescein diacetate is superior for detecting intracellular oxidants: comparison with 2′,7′-dichlorodihydrofluorescein diacetate, 5 (and 6)-carboxy-2′,7′-dichlorodihydrofluorescein diacetate, and dihydrorhodamine, Free Radic. Biol. Med., 27(1-2), 146–159, https://doi.org/10.1016/s0891-5849(99)00061-1, 1999.

Hussar, E., Richards, S., Lin, Z. Q., Dixon, R. P., and Johnson, K. A.: Human health risk assessment of 16 priority polycyclic aromatic hydrocarbons in soils of Chattanooga, Tennessee, USA, Water Air Soil Pollut., 223(9), 5535-5548, https://dx.doi.org/10.1007%2Fs11270-012-1265-7, 2012.

Ikeda, Y., Nakano, M., Ihara, H., Ito, R., Taniguchi, N., and Fujii, J.: Different consequences of reactions with hydrogen peroxide and t-butyl hydroperoxide in the hyperoxidative inactivation of rat peroxiredoxin-4, J. Biochem., *149*(4), 443–453, https://doi.org/10.1093/jb/mvq156,2011.

Jeong, M. S., Yu, K. N., Chung, H. H., Park, S. J., Lee, A. Y., Song, M. R., Cho, M. H., and Kim, J. S.: Methodological considerations of electron spin resonance spin trapping techniques for measuring reactive oxygen species generated from metal oxide nanomaterials, Sci. Rep., *6*(February), 1–10, https://doi.org/10.1038/srep26347, 2016.

Keston, A. S., and Brandt, R.: The fluorometric analysis of ultramicro quantities of hydrogen peroxide, Anal. Biochem., *11*(1), 1–5, https://doi.org/10.1016/0003-2697(65)90034-5, 1965.

Landreman, A. P., Shafer, M. M., Hemming, J. C., Hannigan, M. P., and Schauer, J. J.: A macrophage-based method for the assessment of the reactive oxygen species (ROS) activity of atmospheric particulate matter (PM) and application to routine (daily-24 h) aerosol monitoring studies, Aerosol Sci Technol., 42(11), 946–957, https://doi.org/10.1080/02786820802363819, 2008.

Lane, K. B., Egan, B., Vick, S., Abdolrasulnia, R., and Shepherd, V. L.: Characterization of a rat alveolar macrophage cell line that expresses a functional mannose receptor, J. Leukoc. Biol., *64*(3), 345–350, https://doi.org/10.1002/jlb.64.3.345,1998.

Lin, T. J., Hirji, N., Stenton, G. R., Gilchrist, M., Grill, B. J., Schreiber, A. D., and Befus, A. D.: Activation of macrophage CD8: pharmacological studies of TNF and IL-1β production, J. Immun., 164(4), 1783-1792, https://doi.org/10.4049/jimmunol.164.4.1783, 2000.

Miljevic, B., Hedayat, F., Stevanovic, S., Fairfull-Smith, K. E., Bottle, S. E., and Ristovski, Z. D.: To sonicate or not to sonicate PM filters: Reactive oxygen species generation upon ultrasonic irradiation, Aerosol Sci. Tech., 48(12), 1276–1284, https://doi.org/10.1080/02786826.2014.981330, 2014.

Reiniers, M. J., Van Golen, R. F., Bonnet, S., Broekgaarden, M., Van Gulik, T. M., Egmond, M. R., and Heger, M.: Preparation and Practical Applications of 2′,7′-Dichlorodihydrofluorescein in Redox Assays, Anal. Chem., 89(7), 3853–3857, https://doi.org/10.1021/acs.analchem.7b00043, 2017.

Slamenova, D., Kozics, K., Hunakova, L., Melusova, M., Navarova, J., and Horvathova, E.: (2013). Comparison of biological processes induced in HepG2 cells by tert-butyl hydroperoxide (t-BHP) and hydroperoxide (H2O2): The influence of carvacrol, Mutat. Res. Genet. Toxicol. Environ. Mutagen, 757(1), 15–22, https://doi.org/10.1016/j.mrgentox.2013.03.014, 2013.

Sklorz, M., Briedé, J. J., Schnelle-Kreis, J., Liu, Y., Cyrys, J., de Kok, T. M., and Zimmermann, R.: Concentration of oxygenated polycyclic aromatic hydrocarbons and oxygen free radical formation from urban particulate matter, J. Toxicol. Environ. Health Part A, 70(21), 1866-1869, https://doi.org/10.1080/15287390701457654, 2007.

Wang, W., Jariyasopit, N., Schrlau, J., Jia, Y., Tao, S., Yu, T. W., Dashwood, H. R., Zhang, W., Wang, X., and Simonich, S. L. M.: Concentration and photochemistry of PAHs, NPAHs, and OPAHs and toxicity of PM2. 5 during the Beijing Olympic Games, Environ. Sci. Technol., 45(16), 6887-6895, https://doi.org/10.1021/es201443z, 2011.

**Appendix: Revised manuscript in track mode**

[revised manuscript text omitted]

- Equation: $Y = (0.83 \pm 0.08)\,X - (21.49 \pm 76.09)$
- $R^2 = 0.71$

*Figure 8: Comparison of the OP for manual vs. automated operation using ambient PM samples (N=50). Dotted line represents the identity line.*

[Figure]

[Figure]

*Figure 9: Intrinsic OP of 11 transition metals (panel a), 4 quinones and 7 PAHs (panel b), and 6 inorganic salts (panel c). The*
*concentration in the RV was 0.5 μM for all the metals; 0.2 μM for all quinones and PAHs, and 5 μM for inorganic salts, except for*
*NaCl (1 μM). Error bars represent one standard deviation from the average.*

*Table 1: Limit of detection and precision of the instrument obtained through the measurements of field blanks, positive control*
*and ambient PM samples (n = 10)*

| Sample | Unit | Average | Standard Deviation | LoD | CoV (%) |
|---|---|---|---|---|---|
| Field Blank | mg/mL t-BOOH | 5.30 | 0.42 | 1.26 | 7.95 |
| t-BOOH | % ROS response | 684.71 | 111.13 | - | 16.23 |
| Cu (II) | mg/mL t-BOOH | 71.05 | 10.18 | - | 14.33 |
| Zymosan | mg/mL t-BOOH | 18.84 | 7.15 | - | 37.97 |
| Ambient PM sample | mg of t-BOOH/mg of PM | 402.01 | 57.93 | - | 14.41 |